Spatial planning model for optimizing conservation priorities for local community utilization on Arefi Island in the Raja Ampat Marine Protected Area (MPA) Southwest Papua, Indonesia

Darmawan Mulyanto muly023@brin.go.id 1
Simamora Debora Christi 2
Nahib Irmadi 3
Ramadhani Fadhlullah 1
Sutrisno Dewayany 4
Amhar Fahmi 1
Ramdhan Muhammad 1
Safitri Sitarani 1
Sutejo Bayu 3
Arifin Samsul 1
Agus Syamsul Bahri 2
1 Research Center for Geoinformatics, National Research and Innovation Agency of Indonesia (BRIN) , Bogor , West Java , Indonesia
2 Department of Marine Science and Technology, IPB University, Bogor Institute of Agriculture , Bogor , West Java , Indonesia
3 Research Center for Limnology and Water Resources, National Research and Innovation Agency of Indonesia (BRIN) , Bogor , West Java , Indonesia
4 Research Center for Conservation of Marine and Inland Water Resources, National Innovation and Research Agency (BRIN) , Bogor , West Java , Indonesia
Schuster Richard
Electronic publication date: 2025 Apr 29
Publication date: 2025
Volume: 13
Electronic Location ID: e19292
Received 2024 May 6; Accepted 2025 Mar 19
Copyright: ©2025 Darmawan et al.
Copyright year: 2025
Copyright holder: Darmawan et al.
License: This is an open access article distributed under the terms of the Creative Commons Attribution License, which permits unrestricted use, distribution, reproduction and adaptation in any medium and for any purpose provided that it is properly attributed. For attribution, the original author(s), title, publication source (PeerJ) and either DOI or URL of the article must be cited.
License URL: https://creativecommons.org/licenses/by/4.0/

Keywords: Spatial planning, remote sensing, Marine protected area, indigenous management, Marxan model

Funding: Decision Support System Prototype Based on Satellite Image Analysis Research Fund National Research and Innovation Agency of Indonesia B-11046/III.6/TK.01.00/11/2023 This work was supported by the Decision Support System Prototype Based on Satellite Image Analysis Research Fund (Batch 1) Fiscal Year 2024, National Research and Innovation Agency of Indonesia (Project No.: B-11046/III.6/TK.01.00/11/2023). The funders had no role in study design, data collection and analysis, decision to publish, or preparation of the manuscript.

==============================
This study investigates the application of remote sensing technologies to identify the biophysical characteristics of marine ecosystems for spatial planning, focusing on optimal conservation scenarios within the Raja Ampat Marine Protected Area (MPA) on Arefi Island, Southwest Papua, Indonesia. Indigenous communities manage this area. WorldView-3 satellite imagery, combined with an object-based image analysis (OBIA) approach, was used to classify and map coastal ecosystems. A Marine Reserve Design using the Spatially Explicit Annealing (Marxan) model was applied to delineate conservation areas and propose zoning strategies. Based on ecological values (EV), three scenarios were tested to prioritize conservation features while ensuring sustainable ecosystem use. Image analysis revealed that Arefi Island’s coastal ecosystems cover 64.78 hectares, consisting of seagrass beds (45.41%), coral reefs (36.35%), and mangroves (18.24%), with a kappa accuracy of 0.82. Results indicate that EV3 selects the highest number of planning units, ensuring broader conservation coverage, while EV1 selects the fewest. EV2 is the most budget-friendly option with the lowest cost, whereas EV3 is the most expensive. Ecological Scenario II provided a balanced approach, allocating larger areas for local community use while preserving conservation integrity. Moreover, sensitivity analysis confirmed that a conservation objective targeting 40% of the total area (EV II) is the most effective model for Arefi Island. The zoning breakdown under this scenario includes a core zone of 19.53 hectares, a utilization zone of 15.96 hectares, a sustainable fisheries zone of 15.67 hectares, and other zones covering 92.89 hectares. This study highlights the effectiveness of remote sensing and spatial planning tools, such as Marxan, in marine conservation within indigenously managed areas, emphasizing the importance of balancing conservation efforts with sustainable community use for future planning.

Introduction

Indonesia harbours some of the world’s richest marine biodiversity. It encompasses approximately 3,953,800 hectares of the world’s coral reefs, over 3,000,000 hectares of seagrass beds, and 2,332,429 hectares of mangroves (Amkieltiela et al., 2022; Burke et al., 2011; Hamilton & Friess, 2018; Thorhaug et al., 2020). These productive ecosystems provide numerous benefits, such as filtering pollutants, supplying nutrition, offering coastal protection, supporting livelihoods, and sequestering carbon. Due to these significant benefits, Indonesia has a high conservation priority, particularly in the Raja Ampat Marine Protected Area, located in the Raja Ampat Regency, Southwest Papua Province. While other regions like Komodo National Park, Wakatobi, and Bunaken hold significant conservation value in Indonesia, Raja Ampat is uniquely distinguished by its extraordinary biodiversity, strategic location within the Coral Triangle, and globally significant ecosystems. The unique marine life, ecological significance, and need for sustainable management in the face of growing environmental pressures position Raja Ampat as a top conservation priority for Indonesia. The nation recognizes that safeguarding Raja Ampat is a national obligation and essential for protecting global marine biodiversity. Conservation efforts in Raja Ampat have also focused on supporting the livelihoods of indigenous and local communities (Sutton, 2023).

The conservation of marine environments is crucial for maintaining the Earth’s natural processes, addressing significant challenges like climate change, and promoting societal well-being and benefits (Marcos et al., 2021). In Indonesia, the government has established 411 marine protected areas (MPAs) across its archipelago, covering approximately 9% of its territorial waters—over 28 million hectares (Estradivari et al., 2022). Protecting 30% of these key habitats aims to balance conservation efforts with the sustainable use of marine resources, ensuring ecological resilience while supporting the livelihoods of local fishery-dependent communities (Waldron et al, 2020). Among these, the Raja Ampat marine area was officially established as an MPA under Ministerial decree no. 36 in 2014, issued by the Ministry of Maritime Affairs and Fisheries (MMAF) of the Republic of Indonesia. Spanning approximately 1,026,540 hectares, this MPA is divided into five regions, each incoporating specific zones to support effective conservation sustainable development (MMAF, 2014). In this framework, “zone” refers to a designated area with specific characteristics or function, often intended for conservation, resource management, or regulatory purposes. Raja Ampat zoning system include core zones, utilization, fisheries, and other zones. The “other zones” category is further divided into two subzones: traditional use and seasonal closure, and other utilization (MMAF, 2016).

MPAs are defined as marine, coastal, or small island areas that are protected and managed by a zoning system to achieve the sustainable management of fish resources and biodiversity conservation (Green et al., 2009). MPAs are essential tools for conserving marine biodiversity and sustaining ecosystem services (Claudet et al., 2020). MPAs that incorporate well-designed spatial planning strategies are more effective in achieving conservation goals (Edgar et al., 2014). With the rising threats from human activities—such as overfishing, habitat destruction, and climate change- spatial planning within MPAs has become vital for sustainable management (Mora & Peter, 2011). In area like Arefi Island, these challenges are compounded by sea level rise, ocean acidification, and intensified anthropogenic pressures. Overfishing and habitat destruction remain significant concerns, underscoring the need for effective spatial planning to mitigate these challenges is essential (White et al., 2014). Advanced modelling techniques have positioned spatial planning as a powerful tool for optimizing conservation efforts within MPAs. This approach systematically allocates marine areas for specific purposes, considering ecological, social, and economic goals to ensure sustainable use and long-term biodiversity protection.

The Raja Ampat Regency in Southwest Papua, Indonesia, is renowned for its natural tourism- both on land and at sea- and its rich sociocultural heritage, largely attributed to its extraordinary marine biodiversity (Cinner et al., 2018). However, despite its global importance, the Raja Ampat MPAs face notable challenges, particularly in areas like Arefi Island. Arefi Island and its surroundings are located within the “other zones” of the Raja Ampat marine conservation area (MMAF, 2014). While the importance of Arefi in the broader context of marine conservation is recognized, the current MMAF decree lacks the specificity required to fully harness its potential. In particular, the decree fails to clearly delineate the boundaries of subzones, address the diverse ecosystems and biodiversity within the area, and optimize the use of marine resources by indigenous communities. This lack of precision hinders the realization of conservation objectives. For an MPA to be both effective and beneficial to surrounding communities, its location must adhere to four key principles: Connectedness, Adequacy, Representativeness, and Effectiveness (CARE) (Ban et al., 2011).

Effective maritime planning is vital for designing robust marine conservation strategies. Spatial analysis plays a key role in optimizing decision-making, particularly when budgets are limited. In such case, prioritizing conservation areas with lower socioeconomic costs is essential to meet ecosystem service (ES) (De Groot et al., 2022; Schröter & Remme, 2016). Tools like Marxan are designed to incorporate these costs, enabling cost effective conservation planning to ES targets (Adame et al., 2015; Watson et al, 2019). By accounting for conservation cost as spatial constraints, these tools help prioritize areas where objectives can be achieved at the lowest-cost (Naidoo & Ricketts, 2006; Egoh et al., 2011). Conservation prioritization based on systematic conservation planning (SCP) theory enables cost-effective efforts while addressing multiple objectives (Beger et al., 2022). Originally foundational in conservation biology (Margules & Pressey, 2000), SCP now guides decision-making for prioritizing conservation actions (Kukkala & Moilanen, 2017). SCP priorizes areas based on three criteria: importance, vulnerability, and feasibility, ensuring comprehensive coverage and balanced objectives (Wilson, Carwardine & Possingham, 2009; Kukkala & Moilanen, 2017). Moreover, SCP addresses two key challenges—minimizing costs and maximizing benefits—by providing effective solutions for both planning and implementation (Alagador, Cerdeira & Araújo, 2016).

Integrating maritime planning into a Geographic Information System (GIS) framework provides significant advantages for conservation efforts. This approach enables the evaluation of objectives, identification of marine use conflicts or synergies, risk assessment of human activities, spatial zone management, and scenario testing. Tools such as risk assessments, forecasting, modeling, and simulation models play a pivotal role in supporting efficient conservation planning and addressing complex ecological and management scenarios (Stelzenmüller et al., 2013). This study proposes a novel methodology for determining Other Effective Area-Based Conservation Measures (OECM) by combining Marxan and SCP theory within a GIS framework. By combining these tools, the methodology aim to enhance decion-making processes and optimaize conservation outcomes.

Establishing MPAs is an important step toward conservation, but the lack of precise boundary delineation and inadequate attention to the intricate mosaic of ecosystems hampers the strategic planning needed for effective conservation. This gap in applying CARE principles in Arefi Island’s designation highlights a critical issue in marine conservation efforts in Indonesia and other similarly biodiverse regions worldwide. The generalized approach of the MMAF decree fails to address the ecological and socioeconomic complexities of Arefi Island, leading to a disconnect between conservation objectives and on-ground realities. This oversight not only compromises the ecological integrity of the protected area but also the livelihoods and cultural heritage of indigenous communities dependent on these marine resources. Moreover, the failure to align with recommended coverage and core zone area standards for MPAs exacerbates challenges in achieving sustainable conservation outcomes. Addressing these challenges requires a comprehensive, data-driven approach to MPA management that emphasizes spatial planning and community involvement, ensuring both biodiversity conservation and socioeconomic resilience.

This study aimed to examine the complexities of conserving the Arefi subzone by utilizing existing biodiversity elements through remote sensing data, enabling the local community to make optimal use of these resources. The primary objectives were to use remote sensing data to identify biophysical features (mangroves, coral reefs, and seagrass) as input for determining conservation areas on Arefi Island and to propose zoning within the the Raja Ampat MPA to protect biodiversity while supporting the sustainable management of marine resources by the local community. Through this research, we seek to provide valuable insights into marine conservation planning, contributing to the development of a robust and sustainable spatial plan for Arefi Island.

Materials & Methods

Study area

This research was conducted in Area III of the MPA in the Dampier Strait, Arefi Island, Raja Ampat Regency, Southwest Papua (Fig. 1). Raja Ampat Regency is group of islands situated between 2°25′N and 4° 25′S latitude and 130°E to 132°55′ E longitude. The regency covers approximately 6,084.5 km2 and encompasses around 600 islands of various sizes. The Raja Ampat conservation area is renowned for its ecological richness and is a popular tourist destination. It also holds strategic importance for fisheries due to its well-functioning aquatic ecosystem. Although categorized under “other zones” in the Raja Ampat MPAs, which includes four districts in Southwest Papua, satellite data reveal the significant biophysical potential of Arefi Island (McKenna, Gerald & Suer , 2002; MMAF, 2018). The term “other zone” refers to the zoning classification used by MMAF for the Raja Ampat water conservation area.

Figure 1 Study area map in Arefi Island, the Raja Ampat Marine Protected Area (MPA) Southwest Papua, Indonesia.

This research was conducted in Arefi Island, Raja Ampat Regency, Southwest Papua Province. The Raja Ampat MPA in Southwest Papua, Indonesia, encompasses 147 vast and diverse marine ecosystem, including Arefi Island.

Arefi Island, located at 0°47′18.67″S and 130°42′27.72″E, is home to significant marine biodiversity and provides crucial habitats for various species, including corals, fish, and endangered marine mammals (Kovács et al., 2021; Trip et al., 2019). The island’s unique characteristics make it a suitable candidate for various conservation zones, such as core, fisheries, and sustainable utilization areas. In addition to its ecological importance, Arefi Island is home to indigenous cultures that practice the “Sasi” tradition, a customary resource management system deeply rooted in their cultural heritage. The traditional practice of Sasi, implemented by local communities on Arefi Island, plays a key role in ecosystem recovery by regulating resource use, allowing marine habitats to regenerate and preventing depletion due to overfishing. This system involves periodic closures to allow ecosystem recovery and ensure resource sustainability (Sairiltiata, 2023). Under Sasi, indigenous communities impose temporary bans (moratoriums) on the use of marine resources, such as coral reefs and fish, in specific areas for designated periods (Rachma, Mangunjaya & Tobing, 2018). This highlights the need for a comprehensive and nuanced zoning strategy to fully protect and utilize the island’s diverse ecosystems.

Data used

The data used in this research primarily consists of remote sensing data, supplemented by secondary data sources (Table 1). The integration remote sensing data with ground-based observation or secondary data enhances the accurary of the result (Petrou, Manakos & Stathaki, 2011). Specifications of the multispectral bands of the Worldview-3 imagery are detailed in Table 2. The research framework and stages are illustrated in Fig. 2.

Table 1 Types and sources of data used in this research.

Types and sources of data used in this research include both primary and secondary data, comprising remote sensing data and thematic maps.

Type of data	Data	Resolution	Source	Periods	
Primary data	Worldview-3 Image	0.6 m	BRIN	2021	
Base Map	1: 25.000	BIG	2000	
Secondary data	Mangrove Map	1: 25.000	BIG	2021	
MPA Map	–	MMAF (2016)	2016	
	Spatial Plan (TRTW)	1:250.000	Local Gov.	2022	

Table 2 Worldview-3 satellite specifications.

The multispectral bands of the WorldView-3 satellite imagery specification.

Spectral Bands	Coastal: 400–450, Pan: 450–800 nm, Blue: 450–510 nm, Green: 510–580 nm, Yellow: 585–625, Red: 630–690 nm, NIR: 770–1,040 nm	
Spatial resolutions	0.3 m (pan), 1,24 m (VNIR)	
Data quantization	11 bit/pixel Pan and MS; 14 bit/pixel	
Swath width	13.1 km	
Revisit frequency	1 m GSD: <1.0 day
4.5 days at 20° off-nadir or less	
Geolocation accuracy	<3.0 m CE90 (Circular Error of 90%)	
Orbit altitude	617 km	
Orbit type	Sun-synchronous	
Orbit period	97 min	
Revisit time	1 day at 1-metre GSD resolution
4.5 days at 20° off-nadir (0.59 m GSD)	
Notes.

Source: European Space Agency (2024); Choudhury et al. (2021).

Figure 2 Research framework.

The figure shows the flow of research carried out involving biophysical analysis of high-resolution satellite data, spatial analysis using GIS applications, and Marxan models for zoning planning units and discussions.

The first step involved analyzing satellite imagery to map biophysical parameters, including mangroves, seagrass, and coral reefs. These biophysical parameters serve as breeding grounds for numerous fish species with both commercial and ecological significance (Weeks, 2017; Sutrisno, Sugara & Darmawan, 2021) and are integral to conservation efforts. In the second step, the location of biophysical parameters were used as inputs for determining conservation features. Cost features were calculated based on the current usuage, as presented in Table 3.

Table 3 Cost feature and corresponding score for feature conservation.

Cost Features = 1 (all feature conservation)		
Feature conservation	PU score	PU status	
Residential Areas	3	Lockout (area that can’t be included)	
Land use	3	Lockout (area that can’t be included )	
Floating Net Cage	1	Seed = Starting point (not often used)	
Dock	1	Seed = Starting point (not often used)	
Mangrove	0	Default	
Coral reef	0	Default	
Seagrass	0	Default	

Method of biophysical analysis

The biophysical parameters were mapped using high-resolution Worldview-3 satellite images from 2021, provided by the Center for Data and Information, National Innovation and Research Agency (BRIN). The spatial resolution of these images is approximately 0.6 m, allowing detailed analysis and mapping of terrestrial and aquatic ecosystems.

Due to spectral similarities, traditional pixel-based classification methods are limited for biophysical analysis in shallow water areas. To address this, object-based image analysis (OBIA) was employed, which differs from pixel-based methods by using image objects as the basic unit of analysis rather than individual pixels (Hossain & Chen, 2019). OBIA is needed for high resolution or highly variable images, because it is able to group pixels into objects based on spatial and spectral characteristics, thereby increasing classification accuracy (Blaschke, 2010).

OBIA is an iterative process that starts with segmenting satellite images into cohesive and contiguous segments. These image objects are then classified using either supervised or unsupervised approaches (Belgiu & Csillik, 2018). According to Ventura et al. (2018), the OBIA workflow begins with image segmentation, a process based on pixel parameters with similar spectral values. In this study, we used the multi-resolution segmentation (MRS) algorithm to create image objects that minimize average heterogeneity and maximize homogeneity. The three key parameters in the MRS algorithm are shape, compactness, and scale (Darmawan et al., 2022). OBIA analysis was performed using eCognition Developer 64 software.

The segmentation results were then classified using support vector machine (SVM) algorithms, a sophisticated non-parametric classifier widely employed in hyperspectral image classification that operates based on statistical learning theory (Tan et al., 2018). It is designed to seek an optimal decision hyperplane within a high-dimensional space, ensuring optimal separation of classes. SVM consistently performs well in challenging classification scenarios with high-dimensional features, demonstrating its effectiveness even when dealing with a limited number of training samples (Cao et al., 2018). The fundamental concept behind the SVM is to identify a hyperplane that maximizes the margin between distinct classes. This hyperplane is expressed by the following equation (Camps-Valls & Bruzzone, 2009). fx= ∑i=1nαiyiKx,xi+b

where fx: decision function

αi: Coefficients obtained during the training process

yi: class label of training sample xi

K(x, xi): kernel function

b: bias term

The predicted class of the input data point x is determined by fx. If fx>0 then the data point is classified as belonging to one class. If fx<0 then the data point is classified as belonging to another class. The classified data and field and secondary observations were then input into the Marxan software.

Method of priority area conservation

Marxan model principle

Marine Reserve Design Using Spatially Explicit Annealing (Marxan) is software designed to support the systematic design of conservation areas (Ball, Possingham & Watts, 2009). Marxan aids in identifying conservation areas that offer high sustainability value while maintaining relatively low management costs. It operates using a simulated annealing algorithm, which is developed to rapidly achieve optimal results through iterative optimization (Anggraeni et al., 2017).

The Marxan algorithm involves numerous random changes to the protected area system, often involving one million or more iterations. Initially, all changes to the system are accepted, regardless of their impact on the objective function score. As the annealing process progresses, the likelihood of accepting unfavourable changes (those that increase the objective function score) gradually decreases, while the acceptance of beneficial changes (those that decrease the score) becomes more likely. This approach allows the algorithm to converge on a solution that closely approximates the optimal result (Moilanen & Ball, 2009).

The optimal results represent the lowest total cost and are derived using the following equation (Watts et al., 2017): Total Cost= ∑i=1nCost+BLM×∑Boundary+ ∑i=1nSPF×Penalty.

Cost: The combination of socioeconomic values in each planning unit within the selected solution.

BLM: Boundary Length Modifier is a value set by the user and is related to the level of connectivity between planning units. The higher the Boundary Length value, the denser the solution area.

Boundary: The boundary of the selected area.

SPF: Values set by the user and related to the importance of biodiversity target objectives. The higher the SPF assigned to a feature, the more Marxan prioritizes that feature in the solution.

Penalty: Penalty value assigned if biodiversity protection targets are not achieved (optional).

i: Unit ID in the shapefile.

n: Last Unit ID in the shapefile.

The boundary length in the protected area system was measured by counting the number of planning units that border areas outside the protected system. A fragmented protected area system will have a substantial boundary length. Modifying the boundary length or boundary length modifier (BLM) aims to address connectivity issues by assigning a value based on the importance of maintaining a dense protected area network. BLM is crucial because a fragmented system is typically more challenging and costly to manage (Watts et al., 2017).

Marxan models

To meet conservation feature targets, enhance connectivity between areas, and minimize overall management costs for priority zones, we utilized Marxan v.4.0.6. This software is designed to identify priority conservation areas. The analysis was conducted using the QMarxan Toolbox (2.0.1), a plugin for QGIS 3.18.3. The conservation features of Arefi Island include three critical ecosystems: mangroves, seagrass, and coral reefs. These features were identified using a map generated from object-based image analysis. Arefi Island served as the primary study area, with a buffer zone extending from the coastline to encompass all shallow water habitats within this zone. The study area was divided into hexagonal planning units with a side length of 15 m, resulting in 9,531 planning units (PUs) within the area.

Conservation costs were estimated based on land status of the area or region, modified from Wijayanto, Yulianda & Imran (2021) and Watts et al. (2017), and presented in Table 3. Categories include: Resident Area (3), Land Use (3), Floating Net Cage (1), and Dock Area (1). Land use refers to the land cover on an island that is not identified as part of marine conservation areas.

This study examines the cost attributes of various human spatial utilization activities within the conservation area. Penalty scores are assigned to each cost attribute based on the significance of the activity, following Watts et al. (2017). Higher penalty scores indicate greater difficulty in designating the area as a core conservation zone. For instance, a penalty score of one is assigned to activities such as docks and floating net cages, while higher scores are given to land use and residential areas. These scores reflect the challenge of considering or reclassifying the area as a core zone (Wijayanto, Yulianda & Imran, 2021).

Both conservation features and cost attributes are assigned to each planning unit (PU) without normalization, ensuring that each PU contains values for both conservation features and costs. We then calibrated the Species Penalty Factor (SPF) and the BLM. The SPF was calibrated to appropriately scale the penalty for missing conservation features relative to one another. The BLM was adjusted to identify the optimal value that balances area compactness with cost. As the BLM value increases, the algorithm tends to favor a ‘single large’ design over ‘multiple small’ designs, thereby enhancing connectivity.

Agnew, Fryirs & Leishman (2024) highligted the practicality of using Marxan as an accessible tool to address complex prioritization challenges and to model landscape-scale rehabilitation scenarios over time. Similarly, Chan, Hoshizaki & Klinkenberg (2011) demonstrated that a 50% protection scenario effectively stabilized Marxan solutions for ecosystem services, while Delavenne et al. (2012) found that a 50% conservation target offers stronger ecosystem protection. This threshold is designed to provide optimal protection and ensure ecological sustainability. A comparison of this study’s findings with previous research reveals both advances in conservation planning and the ongoing need for refined spatial analysis in MPAs. For instance, Jones & De Santo (2016) emphasized the importance of integrating both ecological and social data to achieve biodiversity conservation and community benefits when designing the effective MPAs. While Jones & De Santo (2016) focused on balancing ecological and socioeconomic factors, our study concentrated on ecological values and their spatial distribution. This difference highlights the importance of a holistic approach that considers both ecological integrity and human well-being, suggesting that future studies should incorporate more comprehensive socioeconomic analyses to better align conservation efforts with community needs.

Therefore in this research, three EV scenarios were analyzed using QMarxan. Scenario EV I, aligned with Target 3 of the post-2020 Global Biodiversity Framework and IUCN guidelines, aims to protect 30% of identified conservation targets, including coral reefs, seagrass, and mangroves. This scenario is designed to enhance connectivity within the conservation area (IUCN, 2008) while supporting fisheries in the surrounding regions (Firmansyah et al., 2018).

Scenario EV II set a 40% protection target, following the recommendations of Noss et al. (2012) and consistent with Aichi Target 11 (Harris & Holness, 2023) which adopt targets of 10%, 30%, 40%, or 50%, for nature conservation to achieve biodiversity goals.

Aichi Target 11 emphasizes the need for protected areas and other effective conservation measures across geographic regions, including strictly protected zones as well as areas where sustainable use is permitted, as long as species, habitats, and ecosystem functions are adequately protected.

Scenario EV III adopted a 50% conservation target, following the ’Half-Earth’ concept, which advocates protecting 50% of conservation targets. This ambitious scenario aligns with the ecoregional approach proposed by Dinerstein et al. (2017), which seeks to preserve 50% of the terrestrial biosphere for global ecological heritage conservation.

The irreplaceability of each planning unit was measured based on the frequency with which it was selected across 1,000 iterations, with values ranging from 0 to 1,000. Units with higher irreplaceability scores were considered more important for conservation. Planning units scoring between 750 and 1,000 were designated as Priority I, indicating their critical importance for conservation. Units scoring between 500 and 750 were categorized as Priority II, while those with scores between 250 and 500 were labelled as Priority III. Units with scores between 0 and 250 were classified as Priority IV. Any unit with a score of zero was considered a nonpriority zone. Priority I areas were designated as core conservation zones, Priority II areas were allocated for tourism, Priority III areas were identified as fisheries zones, and Priority IV areas were set aside for other uses, such as coastal development. Next, the Priority I areas map (core conservation zones) is overlaid with the biophysical feature areas to identify important habitats.

We validated the outputs of the Marxan model by comparing the conservation areas generated with actual field conditions to ensure that the target species and ecosystems were present in the identified priority areas. This validation was essential to confirm the feasibility of implementing Marxan’s recommendations in the field. Additionally, we refined the model through iterative adjustments, such as modifying the SPF and testing various scenarios. This iterative process allowed us to develop a more robust and optimal conservation strategy.

We simulated three scenarios (30%, 40%, and 50%) with identical costs. Using the output from the Marxan operation, after 1,000 iterations for each scenario, we calculated: (1) the total number of selected planning units, (2) conservation costs, and (3) boundary length (BLM). Based on these results, we calculated conservation cost efficiency, defined as the number of planning units per unit of cost. A higher efficiency value indicates a more suitable scenario (Zhang & Li, 2022).

Conservation cost efficiency is calculated as the ratio of selected planning units to total conservation costs, serving as a metric to evaluate resource utilization in achieving conservation objectives. This method evaluates resource efficiency in achieving conservation goals by running Marxan simulations with multiple iterations (e.g., 1,000) across predefined conservation targets (e.g., 30%, 40%, and 50%). Each iteration generates data on selected planning units, associated conservation costs, and the boundary length modifier. Upon completion, the collected data includes total selected planning units (SPU) and their corresponding conservation costs (CC). Efficiency is calculated using the formula: Conservation Cost Efficiency=Total Selected Planning Units/Total Conservation Costs

This calculation allows for comparing the number of planning units selected per unit of cost across different scenarios. A higher result indicates a more efficient scenario in terms of cost-effectiveness in achieving conservation objectives. By applying this method, we can effectively evaluate and compare the efficiency of different conservation strategies using Marxan.

Results

Satellite image analysis results

Satellite image analysis using OBIA identified three primary coastal ecosystems on Arefi Island: mangroves, seagrasses, and coral reefs. Additionally, the Marxan model incorporates parameters such as floating nets, residential areas, and docks for zoning analysis. These parameters are crucial for determining conservation priorities within the Marxan framework (Fig. 3). The total area of coastal ecosystems on Arefi Island is approximately 64.78 hectares Coral reefs cover 36.35% of this area, making them a significant component. Mangroves and seagrasses are also present but are distributed unevenly across the island. Mangroves, which cover 11.81 hectares (18.24% of the total area), are primarily concentrated in the southeast, while their presence is relatively sparse in residential areas. Seagrass beds, covering 29.42 hectares and constituting 45.41% of the total area, are dominant in the northern part of Arefi Island. The overall classification has a kappa accuracy value of 0.82. Additionally, the presence of a port and floating fish cages indicates local community activities such as shipping, fishing, and tourism.

Figure 3 Ecology map generated from OBIA analysis.

Conservation priority area recommendations for Arefi Island

From the analysis of the maps presented in Fig. 4, three important areas—core zone, utilization zone and sustainable fishery zone—were identified with higher selection percentages. These areas are found in the northern, southeastern, and southwestern waters of Arefi Island. Notably, the eastern part of Arefi Island showed a lack of selected areas for conservation.

Figure 4 The Conservation zone of Arefi Island from the Marxan model as a function of the ecological value scenario.

The proposed conservation zone on Arefi Island under Ecological Value I, 30% (A), Ecological Value II, 40% (B), and Ecological Value III, 50% (C). Each colour indicates zonation. The red colour indicates core zone, the green indicates utilization zone, the blue indicates sustainable fisheries zone, and the grey indicates other zones.

The spatial zoning arrangements for Arefi Island’s conservation areas under the three EV scenarios revealed significant differences in how space is allocated to optimize conservation priorities, as shown in Table 4.

Table 4 Zoning arrangements for the Arefi Island conservation area.

Zone	Conservation scenario	Impact magnitude compared to EV1	
	EV1	EV2	EV3	EV2	EV3	
	(ha)	%	Ha	%	Ha	%	%	%	
Core Zone	12.33	2.27	19.53	3.59	34.37	6.32	58.39	178.75	
Utilization Zone	8.83	1.62	15.96	2.93	11.92	2.19	80.75	34.99	
Sustainable Fisheries Zone	20.05	3.69	15.67	2.88	11.87	2.18	−21.85	−40.80	
Other Zone	502.82	92.43	492.89	90.60	485.87	89.31	−1.97	−3.37	
Total Area	544.03	100	544.03	100	544.03	100			

This study compares three conservation scenarios—Ecological Value I (EV I), Ecological Value II (EV II), and Ecological Value III (EV III)—to evaluate the spatial allocation of core zones and their effectiveness in protecting key habitats, including coral reefs, seagrass beds, and mangroves. The analysis highlights the relationship between the extent of core zones and the level of habitat protection achieved. In EV I, the core zone covers 12.33 hectares, representing only 2.27% of the total area. This scenario provides minimal protection, conserving just 5.82% of coral reefs, 16.25% of seagrass beds, and 29.55% of mangroves (Table 5). A significant portion of the area (92.43%) remains allocated for general use, reflecting limited conservation prioritization.

Table 5 Percentage of key biophysical habitats derived from remote sensing data and overlaid with the core zone under several scenarios.

Scenario core zone	Classification	Areas (ha)	Areas in core zone (ha)	Percentage by key habitats (%)	
Ecological Value I	Coral reefs	23.55
(36.35%)	1.37	5.82	
Seagrass	29.42
(45.41%)	4.78	16.25	
Mangroves	11.81
(18.41%)	3.49	29.55	
Total	64.78	9.64	14.88	
Ecological Value II	Coral reefs	23.55
(36.35%)	5.12	21.74	
Seagrass	29.42
(45.41%)	6.52	22.16	
Mangroves	11.81
(18.41%)	3.49	29.55	
Total	64.78	15.13	23.35	
Ecological Value III	Coral reefs	23.55
(36.35%)	9.62	40.85	
Seagrass	29.42
(45.41%)	12.00	40.79	
Mangroves	11.81
(18.41%)	3.50	29.64	
Total	64.78	25.12	38.77	

EV II introduces an expanded core zone of 19.53 hectares, increasing its share to 3.65% of the total area. This expansion results in improved habitat protection, safeguarding 21.74% of coral reefs, 22.16% of seagrass beds, and 29.55% of mangroves (Table 5). Altogether, this scenario ensures the protection of 23.35% of key habitats, indicating a moderate enhancement in conservation efforts compared to EV I.

EV III represents the most ambitious conservation scenario, with the core zone covering 34.37 hectares, equivalent to 6.32% of the total area. This significant expansion enhances habitat protection dramatically, with 40.85% of coral reefs, 40.79% of seagrass beds, and 29.64% of mangroves included in the core zone (Table 5). In total, 38.76% of key habitats are safeguarded under this scenario, illustrating a strong commitment to conservation priorities.

The progression across the scenarios demonstrates a clear trend toward increasing habitat protection through larger core zones, with EV III achieving the most comprehensive conservation outcomes.

As shown in Table 4, increasing the proportion of protected conservation features across the scenarios leads to a corresponding rise in the number of conservation planning units designated as core and utilization zones. Conversely, it results in a reduction in the areas allocated for sustainable fisheries and other zones. Using the 30% conservation scenario (EV I) as a baseline, expanding the protection targets to 40% (EV II) and 50% (EV III) increased the core zone size by 58.39% and 178.75%, respectively.

However, the increase in utilization zone under EV III (34.99%) was less significant comparedto EV II (80.75%) from EV I. Both EV II and EV III led to a reduction in areas designated for sustainable fisheries and other uses. This shift reflects a deliberate reallocation of spatial zones, with EV III showing a substantial reduction in these areas to accommodate an expanded core zone. This reconfiguration highlights the increased prioritization of conservation as the other zones decrease in size, making room for more core conservation areas.

Table 5 presents the impact of different ecological value scenarios on habitat prioritization within the core zone. Under Ecological Value I, the core zone spans 9.64 ha, with mangroves comprising 29.55%, followed by seagrass (16.25%) and coral reefs (5.82%). In Ecological Value II, the core zone expands to 15.13 ha, increasing coral reef (21.74%) and seagrass (22.16%) coverage, while mangroves remain at 29.55%. Ecological Value III further enlarges the core zone to 25.12 ha, with coral reefs (40.85%) and seagrass (40.79%) becoming dominant, while mangrove coverage slightly declines to 29.64%. This trend demonstrates how ecological priorities influence the spatial distribution of key biophysical habitats.

As the scenarios progress from EV I to EV III, the core zone area increases significantly, incorporating larger proportions of coral reefs and seagrass. While mangrove areas remain constant across all scenarios, their percentage within the core zone decreases as the extent of other habitats expands. Scenario III achieves a more balanced representation of coral reefs and seagrass, whereas Scenario I places greater emphasis on mangroves relative to the core zone.

The results for the multitarget scenario are summarized in Table 6. As shown, increasing conservation targets leads to higher conservation costs and longer boundary lengths, although the pattern of conservation efficiency remains irregular.

Table 6 Comparison of total pu, cost, boundary length and efficiency.

Zone	Conservation scenario	
	EV1	EV2	EV3	
Total number of planning units selected	569	577	849	
Conservation costs	126,975	92,418	161,932	
Boundary length	8,310	9,660	9,900	
Conservation efficiency	0.0045	0.0062	0.0052	

Table 6 compares conservation scenarios based on selected planning units, costs, boundary length, and efficiency. The results reveal trade-offs among these factors. Scenario 3 selects the most units but incurs higher costs, while Scenario 2 is the most efficient (0.0062), indicating optimal resource allocation. Scenario 1 has the lowest efficiency (0.0045). These findings highlight the importance of balancing cost, spatial coverage, and efficiency in conservation planning.

The analysis of costs, length, and efficiency across the scenarios revealed that efficiency does not consistently correlate with cost. Scenario 2 achieves the highest efficiency despite having moderate unit count and costs, indicating a more effective allocation of resources compared to Scenarios 1 and 3. Conversely, Scenario 3 incurs the highest cost but does not deliver proportionally higher efficiency, suggesting diminishing returns as resource investment increases.

Discussion

The biophysical parameters detected from this study resulted in an accuracy of 82% (kappa = 0.82). According to Ventura et al. (2018) and Darmawan et al. (2022), overall accuracy of 82% is quite accurate in shallow marine ecosystems. These three biophysical (coral reefs, mangroves and sea grasses) parameters play an important role in determining conservation zones in Arefi island. Mangroves and seagrasses on Arefi Island regulate sediment runoff, preventing excessive accumulation that could harm coral reefs. These ecosystems form an interconnected network, where mangroves and seagrasses filter sediment from land, protecting coral reefs from sedimentation. In turn, coral reefs buffer wave energy, shielding mangroves and enhancing coastal stability. This mutual support highlights the importance of integrated conservation efforts.

Golbuu et al. (2008) reported similar impacts on coral reef communities exposed to muddy river discharge in Pohnpei, highlighting the interconnected nature of coastal ecosystems. Mangroves, in turn, provide critical benefit from reduced wave impact and fostering mutually beneficial biological connections with coral reefs. This interdependence underscore the importance of preserving coral reefs, seagrass, and mangrove as a cohesive ecological unit. Efforts must focust on minimizing degradation from anthropogenic activities to ensure the resilience and sustainability of thes vital ecosystems.

The OBIA method leverages high-resolution satellite imagery to assess habitats within shallow marine ecosystems. The result from OBIA for identifying biophysical parameters highlight the potential of combining remote sensing data with ground-based observations to improve the accuracy of monitoring efforts. Our study found that spatial planning models effective in identifying optimal conservation priority zones on Arefi Island for local community use within the Marine Protected Area. This approach aligns with Estradivari et al. (2022), who promoted OECMs under draft Target 3 of the Post-2020 Global Biodiversity Framework, which seeks to conserve 30% of marine areas by 2030. OECM recognize and support conservation efforts that extend beyond designated Marine Protected Areas.This finding is consistent with Halpern et al. (2019), who showed that spatial planning model effectively integrates ecological data, habitat suitability assessments, and stakeholder input to identify areas of high conservation value and vulnerability.

Applying this approach will enhance the effectiveness of conservation measures and ensure the long-term sustainability of marine ecosystems within the Raja Ampat MPA. Smaller conservation areas with well-defined boundaries improve management and monitoring capacity, enabling MPAs to better conserve, enhance, and restore the marine environment (Henneberg, 2023). Moreover, transparency in decision-making and active community involvement are also essential for the long-term success of MPAs (Henneberg, 2023).

In our results, the absence of selected conservation areas in the eastern waters of Arefi Island (Fig. 4) is likely due to the lower biodiversity, limited high resulition satellite data, and the cumulative impacts of anthropogenic activities, such as tourism, shipping routes, and aquaculture practices. These impacts are more pronounced in the western and southern waters, where conservation priorities are also affected.

Among the three ecological values (EV), EV II and EV III were the closest to the international standard for conservation scenario. According to Green et al. (2014), marine sanctuary areas should cover 20–40% of each primary habitat to optimize benefits for fisheries management and biodiversity conservation, particularly in the context of climate change. Additionally, the core zone of a marine conservation area should encompass 20–30% of the total area to ensure the sustainability of key biological stocks (Krueck et al., 2017). Indonesia’s Minister of Marine Affairs and Fisheries Regulation No. 31 of 2020 on conservation area management stipulates that the core zone of a conservation area classified as a park must cover at least 10% of the ecosystem or habitat of the target species. The protection targets set in EV II meet the regulatory standards, as well as the guidelines outlined by Green et al. (2014) and Krueck et al. (2017).

The study’s findings on Arefi Island, where a 40% protection target was applied, revealed a conservation area covering 20–30% of the region. This result aligns with previous studies, such as Suprianto, Agus & Arhatin (2018) in the Thousand Islands, Jakarta and Anggraeni et al. (2017) in the coral triangle of Southeast Sulawesi. These earlier studies also identified potential zones for conservation, utilization, and sustainable fishing within marine protected areas (MPAs). Specifically, conservation targets for these habitats in the previous studies were set at 30%, 40%, and 50%.

The findings of this study align with those Anggraeni et al. (2017), who also identified core and utilization zones in the Sunda Banda Seascape using Marxan analysis. While their conservation targets for these habitats were set at 30%, 40%, and 50%. The core zones accounted for 2% to 13% of the total conservation area. In comparison, our results indicate that core zones comprise 2–6% of the total area. Although this proportion remains below 10%, these zones still require protection. The remaining areas are proposed to be managed by indigenous communities under sustainable development principles. This highlights the importance of allocating larger areas for local community use while maintaining conservation integrity. Indigenous communities should play an active role in regional conservation planning, particularly in the Arefi Islands, to ensure a balance between ecological preservation and sustainable resource utilization. These findings align with the study by Estradivari et al. (2022), which emphasizes the need to empower indigenous communities in managing marine conservation areas outside of designated MPAs, known as OECMs. Their study demonstrates that OECMs have significant potential to support marine area-based conservation in Indonesia, including aiding the Indonesian Government in achieving both national and international conservation targets and objectives. According to MMAF (2014), Sasi benefit local communities by promoting sustainable fishing practices, preserving cultural traditions, and accommodating sustainable tourism activities. In Raja Ampat, Sasi involves opening and closing access to specific areas and regulating certain activities to ensure resources sustainability. Respect for local Sasi regulations is vital to the success of conservation efforts

Our study found that spatial planning models using the Marxan approach effectively identify optimal conservation priority zones on Arefi Island for local community use within the Marine Protected Area. Unlike previous research, our study uniquely emphasizes the allocation of conservation zones that balance ecological preservation with the sustainable resource utilization needs of indigenous communities. This approach not only optimizes conservation outcomes but also aligns zoning recommendations with the socio-economic and cultural requirements of local stakeholders, thereby addressing a critical gap in prior Marxan applications in Indonesia.

Previous studies have primarily demonstrated the effectiveness of Marxan in conservation planning. For example, Aulia et al. (2021) applied Marxan in the PISISI region of Simeulue Island to identify no-take zones, successfully protecting 80% of conservation targets. Similarly, Yusuf, Ampou & Sidik (2008) and Sidik, Yusuf & Ampou (2008) recommended no-take zones covering 30% of Gili Sulat and Gili Lawang to conserve coral reefs, mangroves, and seagrass beds. Meanwhile, Firmansyah (2009) applied IUCN criteria delineated conservation zone in Maratua and Kakaban Islands, covering 20.44–25.27% of the total area of interest.

In contrast, our research introduces a novel perspective by prioritizing local community utilization as a core element of conservation zoning—a factor often overlooked in earlier studies. By incorporating this focus, our approach not only addresses an important gap but also promote a more and sustainable framework for conservation planning in Raja Ampat Regency, Papua Province, Indonesia. This region retains strong adherence to customary law for managing indigenous communities issues, including marine conservation. As McKenna, Gerald & Suer (2002) highlight, the cultural values of indigenous Papuan communities align well with marine reef conservation, reinforcing the importance of integrating local traditions and practices into conservation strategies.

However, the application of Marxan does not always fully align with IUCN standards, as demonstrated by Wijayanto, Yulianda & Imran (2021) in Southeast Sulawesi. Their study used Marxan to identify core zones under three different scenarios, with the largest zone covering 1,498 hectares—falling short of IUCN criteria. The analysis considered protection levels of 30%, 50%, and a combination of both, resulting in core zone sizes of 751, 1,008, and 1,498 hectares, respectively. While these scenarios met the critical habitat protection threshold of 30%, none encompassed more than 1% of the total conservation area. In contrast, our findings provide valuable insights for managers and stakeholders, offering guidance for core zone designation, spatial planning, and sustainable development strategies.

This study demonstrates that Marxan-based conservation planning can support sustainable fishing practices and preserve marine biodiversity, which are critical objectives for sensitive ecosystems like Raja Ampat. Specifically, the results show that Marxan’s ability to identify core and utilization zones aligns with the ecological objectives of Sasi, prioritizing high biodiversity area and essential habitats for ecological protection and sustainable fish production. This findings align with a report by Rachma, Mangunjaya & Tobing (2018), who emphasized that the existence of local wisdom and culture of Sasi plays an important role in fostering natural resource conservation in the Maluku Islands

The optimal zoning scenarios produced by Marxan emphasize maintaining biophysical factor by safeguarding critical habitats, such as coral reefs, seagrass and mangroves, while allocating zones for sustainable resource use. These findings underscore the potential of Marxan as a complementary tool to traditional conservation practices like Sasi, ensuring a synergy between local ecological knowledge and scientific methodologies to achieve long-term sustainability.

One limitation of this research is its reliance on static ecological data, which may not fully capture the dynamic nature of marine ecosystems or their responses to climate change and human activities. While the spatial resolution and temporal scope of the data were adequate for initial zoning and scenario planning, they may not reflect subtle but significant ecological change over time. Additionally, despite Marxan’s is an effective tool for conservation planning, it has limitations in modeling complex human–environment interactions. This highlights the need for integratrating more adaptive and participatory planning tools that can respond to evolving ecological and social contexts.

Conclusions

Satellite image analysis using OBIA was successfully employed to map the three key coastal ecosystems on Arefi Island: mangroves, seagrasses, and coral reefs—providing crucial biophysical data to support spatial planning. The primary focus of this study was on the use of remote sensing as a methodological tool to generate accurate and reliable input data for conservation planning, particularly in areas with limited accessibility.

Following the principles of systematic conservation planning, this study applied a straightforward remote sensing approach to map mangrove, coral reef, and seagrass ecosystems. This remote sensing data analysis underscores the importance of strategic spatial planning to balance ecological protection with other land-use demands. The Marxan model was then used to analyze multi-target scenarios and identify priority areas for MPAs. The accuracy of this mapping was supported by a kappa value of 0.82, indicating high classification reliability.

The total coastal ecosystem area of Arefi Island is approximately 64.78 hectares, encompassing coral reefs, seagrass beds, and mangroves. Coral reefs cover 36.35% of the total area, primarily surrounding the island. Seagrass beds, which dominate the northern part of the island, cover 29.42 hectares or 45.41% of the total area. Mangroves occupy 11.81 hectares or 18.24%, mainly concentrated in the southeastern region, with smaller patches found in residential areas.The presence of a port and floating fish cages indicates active shipping, fishing, and tourism by the local community. This study highlights the need to preserve these ecosystems and minimize degradation caused by human activities to maintain the island’s ecological balance and long-term conservation goals.

Our analysis highlighted that a conservation objective targeting 40% of the total area (EV II) is the most effective model for Arefi Island. The zoning breakdown under this scenario includes a core zone of 19.53 hectares, a utilization zone of 15.96 hectares, a sustainable fisheries zone of 15.67 hectares, and other zones covering 92.89 hectares. This approach underscores the importance of incorporating traditional knowledge and community participation into conservation strategies. A balance must be achieved between conservation areas and the marine management rights granted to Indigenous communities to support their livelihoods. This research does not aim to maximize the size of conservation areas but rather to identify an optimal management framework that ensures both effective conservation and sustainable resource use for Indigenous communities.

Integrating traditional conservation practices, such as Sasi, with scientific planning tools like Marxan strengthens marine conservation efforts. This synergy ensures sustainable resource use while maintaining ecological integrity, highlighting the importance of community involvement in conservation planning. These findings advocate for adaptive management strategies and underscore the vital role of geospatial technology in protecting marine biodiversity while supporting sustainable resource use in Indonesia’s coastal ecosystems.

Supplemental Information

Supplemental Information 1 OBIA raw data from Worldview data satellite

Supplemental Information 2 Final analysis of Arefi Island map

Supplemental Information 3 Final scenario data

Supplemental Information 4 Final scenarios from Marxan model

Supplemental Information 5 Final scenario results from Marxan analysis

We would like to express our sincere gratitude to the Center for Data and Information, the Research Center for Geoinformatics, the Research Center for Conservation of Marine and Inland Water Resources, the Research Center for Limnology and Water Resources, the National Research and Innovation Agency (BRIN) for their provision of data, resources, and collaborative efforts that greatly contributed to the continuity of this study in achieving the project’s goals. We also like to thanks the editor and reviewers for their generous and constructive comments on the manuscript that greatly help us to improve the quality of the manuscript.

Additional Information and Declarations

Competing Interests

Author Contributions

Data Availability

The authors declare there are no competing interests.

Mulyanto Darmawan conceived and designed the experiments, performed the experiments, analyzed the data, authored or reviewed drafts of the article, tim leader and writing original draft, and approved the final draft.

Debora Christi Simamora conceived and designed the experiments, analyzed the data, prepared figures and/or tables, writing original draft, and approved the final draft.

Irmadi Nahib performed the experiments, analyzed the data, prepared figures and/or tables, authored or reviewed drafts of the article, writing - review and edit, and approved the final draft.

Fadhlullah Ramadhani performed the experiments, analyzed the data, prepared figures and/or tables, data curation and software programmer, and approved the final draft.

Dewayany Sutrisno analyzed the data, authored or reviewed drafts of the article, english translation check and supervision, and approved the final draft.

Fahmi Amhar analyzed the data, authored or reviewed drafts of the article, supervision and turnitin chek, and approved the final draft.

Muhammad Ramdhan analyzed the data, authored or reviewed drafts of the article, data curation and supervision, and approved the final draft.

Sitarani Safitri performed the experiments, authored or reviewed drafts of the article, data curation, and approved the final draft.

Bayu Sutejo analyzed the data, authored or reviewed drafts of the article, supervision, and approved the final draft.

Samsul Arifin analyzed the data, prepared figures and/or tables, and approved the final draft.

Syamsul Bahri Agus analyzed the data, authored or reviewed drafts of the article, and approved the final draft.

The following information was supplied regarding data availability:

The data used to develop and validate the Boundary Length Modifier (BLM) and Species Penalty Factor (SPF) in this study were processed using the Marxan application.

The processed input datasets, parameter files, and output results are available at Zenodo: Ahli Geospasial. (2025). AhliGeospasial/spatial-planning-model: Data 0.4 (Version datapublished01). Zenodo. https://doi.org/10.5281/zenodo.15005722.

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
