# Peer review of "Spatial planning model for optimizing conservation priorities for local community utilization on Arefi Island in the Raja Ampat Marine Protected Area (MPA) Southwest Papua, Indonesia"

_PeerJ, doi:10.7717/peerj.19292_

## Round 0.1 · original submission · Major Revisions

The authors have written a paper that could fill a gap in spatial conservation planning in a highly diverse marine system, but I agree with the two reviewers that major revisions are required. Both reviewers note that the methods lack clarity and that a large amount of detail should be added to this section to make it replicable. The results can thus not be fully evaluated with a detailed and clear methods section.

Both reviewers also note an abundance of unclear and repetitive language throughout the manuscript. I suggest the authors conduct a thorough proof-read of the manuscript, removing repetitive sections and improving the flow and grammar of the text. As one reviewer notes, the discussion should be focused on the results of the manuscript as well. The title of the manuscript is potentially missing a few words as well - for example, I suggest adding "a" for "utilization in a Marine Protected Area...".

I ask that the authors carefully review each of the suggestions from the two reviewers, including their major and minor comments, and the annotated PDFs the reviewers provide. I think with substantial reworking of the text and clarifying the methods and results will lead to a nicely improved manuscript.

Reviewer 1 ·

Basic reporting

Your introduction needs more detail and clarification. Some references were not cited correctly, for instance, in lines 52, 65, and 70 (check PDF for more details). I suggest that you improve the citations. Moreover, your introduction needs more detail. I suggest you improve the description in lines 97 and 111 - to provide more justification.

The English language should be improved to ensure that an international reader can clearly understand your text. Some examples where the language could be improved include lines 27-30 (you could split this sentence). I suggest you have a colleague proficient in English and familiar with the subject matter review your manuscript or contact a professional editing service.

The figures are not clearly presented in color or legend. I suggest using some references on coloring (check color brewer).

Experimental design

The research question was not clearly written and had inexplicable aims. The authors wrote, "This study aims to explore the complexities of the Arefi subzone's conservation by leveraging existing biodiversity elements so that the community can optimally utilize them. This study aims to redesign zoning within the MPA to protect biodiversity and support the sustainable management of marine resources by the community."

Despite that, the method was written well, with detailed information provided sufficient to replicate.

Validity of the findings

I comment on the authors' extensive steps in running the program and obtaining the results. However, the results were ambiguous because the authors combined them with other studies, for instance, lines 331, 334, and 362. The discussion is in different sections.

The authors also often write down the same sentences, for instance, lines 406-407, 426-428, and 432-434. Moreover, the authors need to elaborate more on the results instead of repeating the same sentences.

Additional comments

no comments

Annotated reviews are not available for download in order to protect the identity of reviewers who chose to remain anonymous.

·

Basic reporting

The authors aim to fill a critical gap in MPA spatial planning for a region of important biodiversity. They apply novel and relevant methods and have undertaken a significant amount of work. Their introduction provided the necessary background information and sets up the justification of their study well. Unfortunately the languages is unclear and ambiguous in many critical places in this manuscript. In a number of places sentences and entire paragraphs have simply been copy and pasted. This results in the paper being extremely repetitive and whole sections are uninformative.
Majority of the tables are unnecessary and do not add value to the manuscript. The figures could be combined and improved to better present the background and results of this study.
Raw (or any) data is not in the supplemental file as indicated by the authors.

Experimental design

The experimental design of this study is unclear and the methods do not provide sufficient detail to replicate this experiment. For example, I know the authors compared three scenarios with varying percentages (30%, 40%, and 50%), but I do not know what those are percentages of. The introduction made it seem like they intended to redesign the zones of the MPAs in question but we were not told what or where the current zones were. Instead the aim/research question needs to be rephrased and more clearly defined.

Validity of the findings

I can not comment on the validity of main findings since the methods are unclear.
The information provided about the underlying models and software seem valid.
No data were provided that I can find, the article also doesn't mention a repository.

Additional comments

I have provided in-line comments on the attached pdf.

The most pressing issue of this manuscript is improving the clarity of the methods.

I believe the authors have the makings of an interesting paper, but it is not there yet.

---

## Round 0.2 · Major Revisions

Both reviewers and I agree that substantial improvements have been made to the manuscript, some work is still needed before the paper can be accepted. In particular, some results are only mentioned in the discussion (the sensitivity analysis), and all results should be presented in the Results section prior to discussion. The discussion needs to focus on the authors' results, citing relevant literature, but not act as a literature review per se. A nicely annotated pdf of the manuscript is included here with detailed comments.

I encourage the authors to address each of the reviewers' comments and questions, which will improve the flow and readability of the paper.

Reviewer 1 ·

Basic reporting

The authors have addressed the feedback effectively, and I have no further comments at this time.

Experimental design

The authors have addressed the previous comment. However, some adjustments are needed:

For lines 312-313:
"The method includes areas with the following designations: Residential Area (score: 1), Land Use (score: 3), Floating Net Cage (score: 1), and Dock Area (score: 3). Please note that the numbers in brackets represent the respective scores, as outlined in Table 3."

For line 316:
"Table 3: Cost Features and Corresponding Scores for Land and Water Use. The table title should provide more detailed information to indicate the contents and purpose of the table."

Validity of the findings

The underlying data are comprehensive and statistically sound. The conclusions are well-formulated, directly address the original research question, and are limited to results that are supported by the data. However, I have some specific comments:

Line 389, Figure 3: The color representing the "Sea" in the legend differs from the color on the map. Additionally, the "Dock" and "Floating Net" are not easily identifiable on the map, which may reduce the clarity of the figure.

Line 423: The findings are inconsistent. The manuscript states that "EV II expands the core zone to 15.13 hectares." However, Table 5 lists the core zone for the EV2 scenario as 19.53 hectares. This discrepancy should be addressed for consistency and accuracy.

Additional comments

No comment

·

Basic reporting

The authors made considerable changes throughout the manuscript. For example the research question and aims are much clearer this round. However the writing could be significantly improved in many places. The introduction should be reorganized to go from more general to specific. The discussion section needs much more work. In many places it reads like a literature review rather than a discussion of the author's work and often it is unclear what study the authors are referring to.

The methods section is improved and the three scenarios are described clearly.
I still believe some of the tables are irrelevant but the figures are much better.

Experimental design

The research question is much better defined in this version and it is clear how the authors believe their work will fill relevant gaps in scientific knowledge.
The methods are described with sufficient detail for the most part. See a few in line comments.

Validity of the findings

While it is clear the authors did a large amount of research the findings of this research are poorly presented.
Some results are mentioned for the first time in the discussion section. For example, it was not clear if they ran sensitivity analyses till the discussion section. It is also not clear if they compared their results of remote sensing to other data, they do present a kappa accuracy value but do not define this in the methods.
Conclusions are better stated than in the first draft but unfortunately still not publication ready.

Additional comments

It is clear the authors have improved their writing but it still is not publication ready. I suggest the authors reread some of the literature in their field paying attention to the structure of the introduction and discussion sections. Concentrate on improving the flow of ideas throughout this section so that the reader can have an easier time following the logic of your results.

---

## Round 0.3 · Minor Revisions

The authors have made substantial progress in revising this manuscript, and while the remaining comments are largely related to style, there are still some parts of the methods that require additional detail to make the study reproducible by others. In particular, the authors do not adequately describe how the species penalty factor (SPF) and boundary length modifier (BLM) are calibrated. Calibrating these parameters requires substantial work, and more details are needed on how this was done, which will ultimately impact the overall Marxan results. Similarly, additional detail is needed in lines 371-373 discussing modifying the SPF to refine the model, rather than simply mentioning that SPF was modified to refine the model.

I also encourage the authors to streamline the Discussion - as it stands, the Discussion is broken into many paragraphs which are about two sentences long. This is somewhat disjointed and difficult to read, and some paragraphs are redundant. During the these next revisions, I think some paragraphs could be merged to increase the flow of the discussion. The conclusions paragraphs in particular, are mainly redundant, and could be shortened and merged to just highlight the key story of this manuscript, and next steps, if any.

I think that this is an important manuscript for marine conservation science, and encourage the authors to provide additional detail (and corresponding results if needed) on the SPF/BLM calibrations, and streamlining the Discussion text. Thanks again to the two reviewers who provided comments.

Reviewer 1 ·

Basic reporting

Figures are already edited well and are now well labelled and described.

Experimental design

no comment

Validity of the findings

no comment

Additional comments

no comment

·

Basic reporting

The authors again have made considerable changes to the writing and presentation of this manuscript. While, I still find multiple paragraphs repetitive and unnecessary this could be interpreted as a writing style choice. I have made comments on some (but not all) the places where this occurs in the annotated document attached.
Authors have made the raw data available.

Experimental design

The research question is much better defined in this version and it is clear how the authors believe their work will fill relevant gaps in scientific knowledge.
The methods are described with sufficient detail for the most part. See a few in line comments.

Validity of the findings

Authors have improved the presentation of their results and they are now much more clear. Few minor comments attached, ensure all terms are defined and methods clearly presented (i.e results match the methods section and vice versa).

Additional comments

At this point the discussion could use some more polishing but these constitute editorial changes.

---

## Round 0.4 · Minor Revisions

The authors have made substantial improvements to the manuscript over several rounds of thoughtful revision, however, I do not see any links to the code used which the authors state is linked throughout their document. I have re-read the full manuscript and do not see these links, so they need to be made obvious. A data availability or open data statement at the end of the manuscript would be helpful, linking to the data and code used for some complex methods such as developing and validating the BLM and SPF for the study.

Several minor revisions still require addressing, such as the following, and can be addressed now and at the proofing stage:

Line 331: why is "Favor" capitalized?
Line 355: this sentence is out of place and not really relevant to the methods; it could be mentioned more in the introduction
Line 640: 'plys' should be plays
Line 640-642 seems out of place and could be mentioned at the very start or end of the conclusion, and included in a paragraph rather than just a single sentence in between paragraphs

If the authors can provide easy access to their data/code for review purposes and address some additional minor grammar/structure issues, I think the manuscript can be accepted for publication.

---

## Round 0.5 · Minor Revisions

I thank the authors and reviewers for several helpful rounds of revision. The authors have included minor text suggestions as well as their Marxan data.

There are however still a number of issues with the data and code:

1. The authors put the information they provided on Google Drive. This is not a permanent place to share this kind of information and could be modified or deleted anytime. Please submit your code and data to a permanent repository such as Zenodo or Figshare where you get a DOI for your submission and submissions are permanent.

2. The readme file of how to use data and code is very short which is not enough for people that are fairly new to the methodology to reproduce results. Please expand that file to make sure interested readers are able to reproduce your results.

3. In the readme file there is the following information:
"D. Documentation (/Documentation/) - Marxan-User-Manual_2021.pdf – Explanation of how BLM and SPF values were chosen and step-by-step guide for reproducing the analysis."

The Authors included the actual Marxan user manual that's been published by others and in that document we find guidance on general ways how BLM and SPF values can be calibrated, but nothing specific about this work.
Please provide information on callibration steps taken that are specific to this work.

Could you please address the points raised above in a revision?

---

## Round 0.6 · accepted · Accept

Thank you very much for making these final edits to your work. I am satisfied with your edits and am happy to accept your manuscript for publication.